

# Relational graph convolutional networks: a closer look

Thiviyan Thanapalasingam[1,2], Lucas van Berkel[1], Peter Bloem[2] and Paul Groth[1]

[1] University of Amsterdam, Amsterdam, Noord Holland, Netherlands
[2] VU University Amsterdam, Amsterdam, Noord Holland, Netherlands

## ABSTRACT

In this article, we describe a reproduction of the Relational Graph Convolutional Network (RGCN). Using our reproduction, we explain the intuition behind the model. Our reproduction results empirically validate the correctness of our implementations using benchmark Knowledge Graph datasets on node classification and link prediction tasks. Our explanation provides a friendly understanding of the different components of the RGCN for both users and researchers extending the RGCN approach. Furthermore, we introduce two new configurations of the RGCN that are more parameter efficient. The code and datasets are available at https://github.com/thiviyanT/torch-rgcn.

## INTRODUCTION

Knowledge Graphs are graph-structured knowledge bases, representing entities and relations between pairs of entities (*Nickel et al., 2016*). They have become critical for large-scale information systems for tasks ranging from question answering to search (*Noy et al., 2019*). The ability to perform statistical relational learning over such data enables new links, properties, and types to be inferred (*Nickel et al., 2016*), and performing this at a large scale is fundamental for the advancement of the Semantic Web. Additionally, models that can be applied to Knowledge Graphs can also be applied to Relational Database Management Systems, because there is a one-to-one mapping between the two (*Bornea et al., 2013*).

Relational Graph Convolution Networks (RGCNs) (*Schlichtkrull et al., 2018*) are message passing frameworks for learning valuable latent features of relational graphs. RGCNs have become widely adopted for combining Knowledge Graphs with machine learning applications (the original article has received over 1,200 citations) for their uses include Knowledge Graph refinement (*Paulheim, 2016*), soft-query answering (*Daza & Cochez, 2020*), and logical reasoning (*Sinha et al., 2020*). The original reference code for the RGCN is built on old platforms that are no longer supported. Furthermore, other reproductions (*Wang et al., 2020*; *Fey & Lenssen, 2019*) are incomplete (At the time of writing, we are aware of additional implementations in PyTorch Geometric (*Fey & Lenssen, 2019*), which reproduces only the RGCN layer, and in Deep Graph Library (*Wang*

Corresponding author
Thiviyan Thanapalasingam,
t.singam@uva.nl

*et al., 2020*), which provides the node classification and link prediction model. However, they are not focused on scientific reproducibility). Existing descriptions of the model are are mathematically dense, with little exposition, and assume prior knowledge about geometric deep learning.

We reproduced the model in PyTorch, a widely-used framework for deep learning (*Paszke et al., 2019*). Using our reproduction, we provide a thorough and friendly description for a wider audience. The contributions of this article are as follows:

1. A *reproduction of the experiments* as described in the original article by *Schlichtkrull et al. (2018)*;
2. A new publicly available PyTorch implementation of the model called *Torch-RGCN*;
3. A description of the *subtleties of the RGCN* that are crucial for its understanding, use and efficient implementation;
4. *New configurations* of the model that are parameter efficient.

The rest of the article is organized as follows. We begin with an overview of the related work in "Literature review". Then, we introduce the RGCN in "Relational graph convolutional network", followed by a description of our re-implementation (Torch-RGCN). We then proceed to describe the reproduction of node classification (Downstream task: node classification) and link prediction models with associated experiments (Downstream task: link prediction). These sections include the model variants mentioned above. In the "Conclusion", we discuss the lessons learned from reproducing the original article and the implications of our findings. Finally, we discuss our results and conclude.

## LITERATURE REVIEW

The ability to reproduce published work is the essence of scientific research. In machine learning research, reproducibility is crucial for three primary reasons: (1) Validating and verifying methods during review cycles, (2) reproducing implementations to be used as baselines, and (3) reproducing and improving previously published work. However, reproducibility of machine learning research methods still remains to be challenging for various reasons, such as modern research outgrowing older methods for communicating research results (*Tatman, VanderPlas & Dane, 2018*) and the lack of rigorous standards for reproducibility in the field (*Heil et al., 2021*). Introducing standards for machine learning reproducibility would make it easier for assessing and comparing which features are required to fully support reproducibility (*Isdahl & Gundersen, 2019*). *Tatman, VanderPlas & Dane (2018)* propose a simple taxonomy for describing reproducibility of machine learning research articles into three categories, where the highest category describes work that provides the code, data, and full computational environment necessary to reproduce the results of the study. Similarly, *Heil et al. (2021)* propose standards for machine learning in the life sciences based on data, model and code publication, programming best practices and workflow automation. Ideally, reproduction is fully automated, often referred to as *out-of-box reproducibility*, allowing researchers to reproduce computational methods with little to no effort (*Isdahl & Gundersen, 2019*; *Heil et al., 2021*).

Machine learning over Knowledge Graphs involves learning low-dimensional continuous vector representations, called Knowledge Graph Embeddings (KGEs). Commonly, KGE models are developed and tested using link prediction benchmark datasets. Two key examples of KGEs are TransE (*Bordes et al., 2013*), in which relations are translations operating on the low-dimensional vector representations of entities, and DistMult (*Yang et al., 2015*) where the likelihood of a triple is quantified by a multiplicative scoring function. Similarly, RGCNs can be used to embed Knowledge Graphs. However, RGCNs are different from traditional link predictors, such as TransE and DistMult, because RGCNs explicitly use nodes' neighborhood information for learning vector representations for downstream tasks (*Battaglia et al., 2018*). Incorporating logical background knowledge into KGE models allow generalisation to relations not seen in the training set (*Donadello & Serafini, 2019*). On the other hand, RGCNs strictly operate under a transductive setting where entities and relations in the test set must be available during training.

Besides RGCNs, there are other graph embedding models for relational data. Relational Graph Attention Networks use self-attention layers to learn attention weights of edges in relational graphs but yields similar, or in some cases poor, performance when compared to the RGCN (*Busbridge et al., 2019*). Heterogenous Information Networks (HINs) exploit meta-paths (a sequence consisting of node types and edge types for modelling particular relationships) to low-dimensional representation of networks (*Huang & Mamoulis, 2017*). HINs do not use message passing and their expressivity depends on the selected meta-paths.

Beyond node classification and link prediction, RGCNs have other practical applications. *Daza & Cochez (2020)* have explored RGCNs for soft-query answering by embedding queries structured as small relational graphs. RGCNs can facilitate zero-shot entity recognition by transferring the knowledge obtained from familiar entities to unfamiliar ones (*Chen et al., 2020*). *Mylavarapu et al. (2020)* have shown that spatial information for dynamic scene understanding can be encoded as relations between objects using RGCNs. RGCNs can complement traditional machine learning approaches by reasoning on contextual information (*Hu et al., 2021*). Furthermore, RGCNs have contributed to the field of natural language processing by improving dependency tree extraction (*Guo et al., 2021*).

In this section, we discussed a few examples of recent works that highlight the prevalence of RGCNs and thus, show that RGCNs are still relevant today. We refer the reader to *Ruffinelli, Broscheit & Gemulla (2020)* and *Rossi et al. (2021)* for a comprehensive survey of the state of the art KGE models, and *Wu et al. (2021)* for an overview of graph representation learning frameworks.

## RELATIONAL GRAPH CONVOLUTIONAL NETWORK

*Schlichtkrull et al. (2018)* introduced the RGCN as a convolution operation that performs message passing on multi-relational graphs. In this section, we are going to explain how Graph Convolution Networks operate on undirected graphs (*Kipf & Welling, 2017*) and

how they can be extended for relational graphs. We will describe message passing in terms of matrix multiplications and explain the intuition behind these operations.

## Message passing

We will begin by describing the basic Graph Convolutional Network (GCN) layer for *directed graphs*[1]. This will serve as the basis for the RGCN in "Extending GCNs for multiple relations".

The GCN (*Kipf & Welling, 2017*) is a graph-to-graph layer that takes a set of vectors representing nodes as input, together with the structure of the graph and generates a new collection of representations for nodes in the graph. A directed graph is defined as $\mathcal{G} = (\mathcal{V}, \mathcal{E})$, where $\mathcal{V}$ is a set of vertices (nodes) and $\langle i, j \rangle \in \mathcal{E}$ is a set of tuples indicating the presence of *directed* edges, pointing from node $i$ to $j$. Equation (1) shows the message passing rule of a *single layer* GCN for an undirected graph, $\mathcal{G}$.

$$H = \sigma(AXW), \tag{1}$$

Here, $X$ is a node feature matrix, $W$ represents the weight parameters, and $\sigma$ is a non-linear activation function. $A$ is a matrix computed by row-normalizing[2] the adjacency matrix of the graph $\mathcal{G}$. The row normalization ensures that the scale of the node feature vectors do not change significantly during message passing. The node feature matrix, $X$, indicate the presence or absence of a particular feature on a node.

Typically, more than a single convolutional layer is required to capture the complexity of large graphs. In these cases, the RGCN layers are stacked one after another so that the output of the preceeding RGCN layer $H^{(l-1)}$ is used as the input for the current layer $H^{(l)}$, as shown in Eq. (2). In our work, we will use superscript $l$ to denote the current layer.

$$H^{(l)} = \sigma\left(A\,H^{(l-1)}\,W\right), \tag{2}$$

If the data comes with a feature vector for each node, these can be used as the input $X$ for the first layer of the model. If feature vectors are not available, one-hot vectors, of length $N$ with the non-zero element indicating the node index, are often used. In this case, the input $X$ becomes the identity matrix $I$, which can then be removed from Eq. (1).

We can rewrite Eq. (1) to make it explicit how the node representations are updated based on a node's neighbors:

$$\mathbf{h}_i = \sigma\left[\sum_{j \in N(i)} \frac{1}{|N(i)|} \mathbf{x}_i^T W\right]. \tag{3}$$

Here, $\mathbf{x}_i$ is an input vector representing node $i$, $\mathbf{h}_i$ is the output vector for node $i$, and $N(i)$ is the collection of the incoming neighbors of $i$, that is the nodes $j$ for which there is an edge $\langle j, i \rangle$ in the graph. For simplicity, the bias term is left out of the notation but it is usually included. We see that the GCN takes the average of $i$'s neighbouring nodes, and then applies a weight matrix $W$ and an activation $\sigma$ to the result. Multiplying $\mathbf{x}_i^T W$ by $\frac{1}{|N_i|}$

[1] The original Graph Convolutional Network (*Kipf & Welling, 2017*) operates over undirected graphs.

[2] For undirected graphs, a symmetrically normalized Laplacian matrix is used instead (*Kipf & Welling, 2017*).

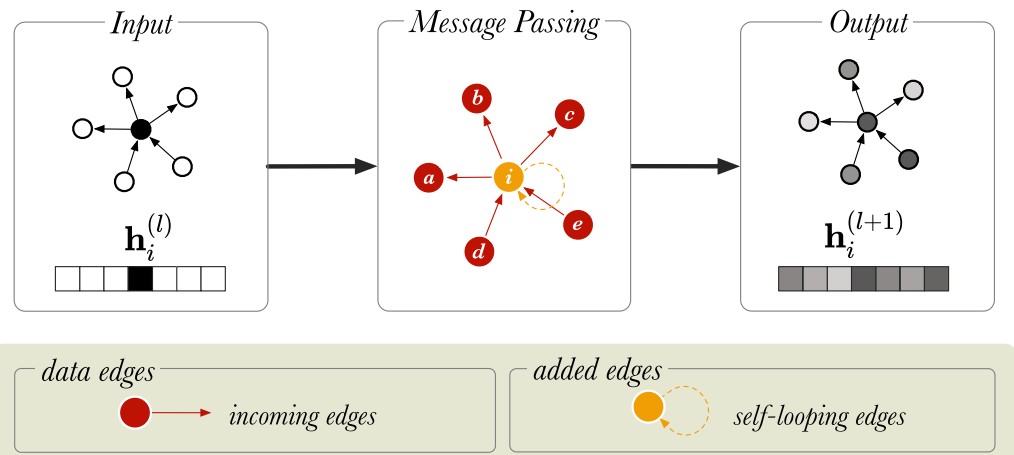

**Figure 1 A schematic diagram of message passing in a directed graph with six nodes.** $h_i$ is a vector that represents the node embedding of the node $i$ (in orange). $h_i^{(l)}$ and $h_i^{(l+1)}$ show the node embedding before and after the message passing step, respectively. The neighboring nodes are labelled from $a$ to $e$.

means that we sum up all the feature vectors of all neighboring nodes. This makes every convolution layer *permutation equivariant*, and that is: if either the nodes (in *A*) are permuted, then the output representations are permuted in the same way. Overall, this operation has the effect of passing information about neighboring nodes to the node of interest, *i* and is called *message passing*. Message passing is graphically represented in Fig. 1 for an undirected graph, where messages from neighboring nodes ($a - e$) are combined to generate a representation for node *i*. After message passing, the new representation of node *i* is a mixture of the vector embeddings of neighboring nodes.

If a graph is sparsely connected, a single graph-convolution layer may suffice for a given downstream task. Using more convolutional layers encourages mixing with nodes more than 1-hop away, however it can also lead to output features being oversmoothed (*Li, Han & Wu, 2018*). This is an issue as the embeddings for different nodes may be indistinguishable from each other, which is not desirable.

In summary, GCNs perform the following two operations: (1) They replace each node representation by the average of its neighbors, and (2) they apply a linear layer with a nonlinear activation function $\sigma$. There are two issues with this definition of the GCN. First, the input representation of node *i* does not affect the output representation, unless the graph contains a self-loop for *i*. This is often solved by adding self-loops explicitly to all nodes. Second, only the representations of nodes that have incoming links to *i* are used in the new representation of *i*. In the relational setting, we can solve both problems elegantly by adding relations to the graph which we will describe in the next section.

## Extending GCNs for multiple relations

In this section, we explain how the basic message passing framework can be extended to relational graphs, such as Knowledge Graphs. We define a Knowledge Graph as a directed graph with labelled vertices and edges. Formally, this graph can be defined as

$\mathcal{G} = (\mathcal{V}, \mathcal{E}, \mathcal{R})$, where $\mathcal{R}$ represents the set of edge labels (relations) and $\langle s, r, o \rangle \in \mathcal{E}$ is a set of tuples representing that a subject entity $s$ and an object entity $o$ are connected by the relation $r \in \mathcal{R}$.

The Relational Graph Convolutional Network extends graph convolutions to Knowledge Graphs by accounting for the directions of the edges and handling message passing for different relations separately. Equation (4) is an extension of the regular message passing rule (Eq. (1)).

$$H = \sigma \left( \sum_{r=1}^{R} A_r X W_r \right), \tag{4}$$

where $R$ is the number of relations, $A_r$ is an adjacency matrix describing the edge connection for a given relation $r$ and $W_r$ is a relation-specific weight matrix. The extended message passing rule defines how the information should be mixed together with neighboring nodes in a relational graph. In the message passing step, the embedding is summed over the different relations[3]. We can rewrite Eq. (4) to show how the node representations are updated based a node's neighbors connected *via* different relations:

$$\mathbf{h}_i = \sigma \left[ \sum_{r}^{R} \sum_{j \in N_r(i)} \frac{1}{|N_r(i)|} \mathbf{x}_i^T W_r \right], \tag{5}$$

where $N_r(i)$ is the collection of the incoming neighbors of $i$ with the relation $r$.

With the message passing rule discussed thus far, the problem is that for a given triple $\langle s, r, o \rangle$ a message is passed from $s$ to $o$, but not from $o$ to $s$. For instance, for the triple $\langle$`Amsterdam, located_in, The_Netherlands`$\rangle$ it would be desirable to update both `Amsterdam` with information from `The_Netherlands`, and `The_Netherlands` with information from `Amsterdam`, while modelling the two directions as meaning different things. To allow the model to pass messages in two directions, the graph is amended inside the RGCN layer by including inverse edges: for each existing edge $\langle s, r, o \rangle$, a new edge $\langle o, r', s \rangle$ is added where $r'$ is a new relation representing the inverse of $r$. A second problem with the naive implementation of the (R)GCN is that the output representation for a node $i$ does not retain any of the information from the input representation. To allow such information to be retained, a self-loop $\langle s, r_s, s \rangle$ is added to each node, where $r_s$ is a new relation that expresses identity. Altogether, if the input graph contains $R$ relations, the amended graph contains $2R + 1$ relations: $\mathcal{R}^+ = \mathcal{R} \cup \mathcal{R}' \cup \mathcal{R}_s$. This is graphically represented in Fig. 2 for a directed graph.

## Reducing the number of parameters

We use $N_{in}$ and $N_{out}$ to represent the input and output dimensions of a layer, respectively. While the GCN (*Kipf & Welling, 2017*) requires $N_{in} \times N_{out}$ parameters, relational message passing uses $R^+ \times N_{in} \times N_{out}$ parameters. In addition to the extra parameters required for a separate GCN for every relation, we also face the problem that Knowledge Graphs do not usually come with a feature vector representing each node. As a result, as we saw in the previous section, the first layer of an RGCN model is often fed with a one-hot vector for

[3] Since RGCN layers are stacked such that the input of a layer is the output of the previous layer, taking the sum over $R$ actually inflates the activations. However, for two-layer networks this does not seem to affect performance. For deeper models, taking the mean over the relations rather than the sum may be more appropriate.

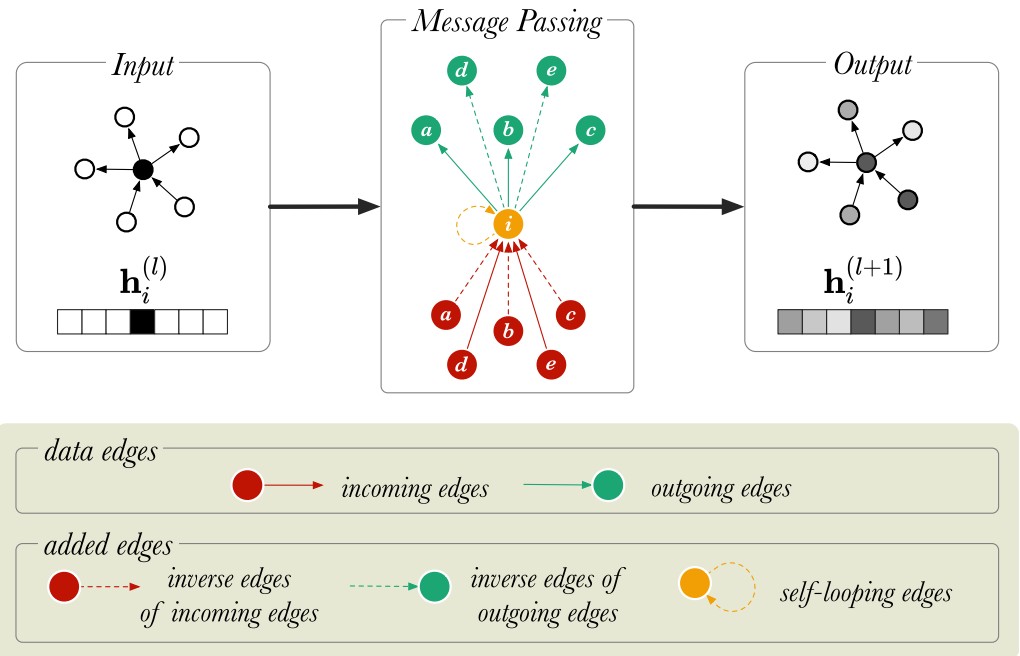

**Figure 2** **A diagram of message passing in a directed, labelled graph with six nodes.** $h_i$ is a vector that represents the node embedding of the node $i$ (in orange). $h_i^{(l)}$ and $h_i^{(l+1)}$ show the node embedding before and after the message passing step, respectively. The neighboring nodes are labelled from $a$ to $e$.

each node. This means that for the first layer $N_{in}$ is equal to the number of nodes in the graph.

In their work, *Schlichtkrull et al. (2018)* introduced two weight regularisation techniques: (1) Basis Decomposition and (2) Block Diagonal Decomposition. We believe that these matrix decomposition techniques help to reduce the number of parameters (*i.e.*, increasing parameter efficiency) and thus, prevent the model from overfitting to the training datasets. Figure 3 shows visually how the two different regularisation techniques work.

**Basis Decomposition** does not create a separate weight matrix $W_r$ for every relation. Instead, the matrices $W_r$ are derived as linear combinations of a smaller set of $B$ basis matrices $V_b$, which is shared across all relations. Each matrix $W_r$ is then a weighted sum of the basis vectors with component weight $C_{rb}$:

$$W_r = \sum_{b=1}^{B} C_{rb} V_b, \tag{6}$$

Both the component weights and the basis matrices are learnable parameters of the model, and in total they contain fewer parameters than $W_r$. With lower number of basis functions, $B$, the model will have reduced degrees of freedom and possibly better generalisation.

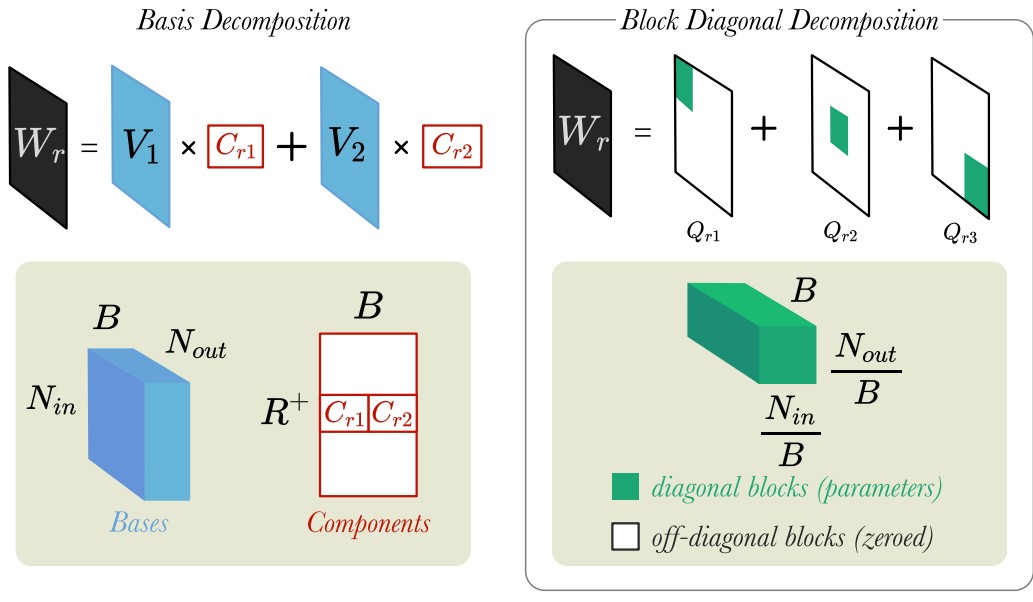

**Figure 3 A simplified visualisation of the weight regularisation methods.** Left: A weight matrix, $W_r$ is decomposed into two bases ($B = 2$). Bases are represented by a tensor $V \in \mathbb{R}^{B \times N_{in} \times N_{out}}$ and components by matrix $C \in \mathbb{R}^{R^+ \times B}$. Right: A different weight matrix, $W_r$, is decomposed into three blocks ($B = 3$). Blocks are tensors $Q_{rb} \in \mathbb{R}^{B \times \frac{N_{in}}{B} \times \frac{N_{out}}{B}}$. $N_{in}$ and $N_{out}$ are the input and output dimensions of layer, respectively.

**Block Diagonal Decomposition** creates a weight matrix for each relation, $W_r$, by partitioning $W_r$ into $\frac{N_{in}}{B}$ by $\frac{N_{out}}{B}$ blocks, and then fixing the off-diagonal blocks to zeros (shown in Fig. 3)[4]. This deactivates the off-diagonal blocks, such that only the diagonal blocks are updated during training. An important requirement for this decomposition method is that the width/height of $W_r$ need to be divisible by $B$.

$$W_r = \begin{bmatrix} Q_{r1} & 0 & \cdots & 0 \\ 0 & Q_{r2} & 0 & 0 \\ \vdots & 0 & \ddots & \vdots \\ 0 & 0 & \cdots & Q_{rB} \end{bmatrix} \tag{7}$$

Here, $B$ represents the number of blocks that $W_r$ is decomposed into and $Q_{rb}$ are the diagonal blocks containing the relation-specific weight parameters. Equation (7) shows that taking the direct sum of $Q_r$ over all the blocks gives $W_r$, which can also be expressed as the sum of $\text{diag}(Q_{rb})$ over all the diagonal elements in the block matrix. The higher the number of blocks $b$, the lower the number of trainable weight parameters for each relation, $W_r$, and *vice versa*. Block diagonal decomposition is not applied to the weight matrix of the identity relation $r_s$, which we introduced in "Extending GCNs for multiple relations" to add self-loops to the graph.

[4] The off-diagonal blocks are $\frac{N_{in}}{B}$ by $\frac{N_{out}}{B}$ matrices containing only zeros. In Eq. (7), we present these zeroed blocks simply with 0.

## TORCH-RGCN

The original implementation of the RGCN by *Schlichtkrull et al. (2018)* was written using two different differentiable programming frameworks, Theano 0.9 (*Al-Rfou et al., 2016*) and TensorFlow 1.4 (*Abadi et al., 2016*)[5]. Both frameworks have become obsolete in the past 2 years. Therefore, we have reproduced the RGCN using PyTorch (*Paszke et al., 2019*). We will refer to our implementation as *Torch-RGCN* and the original implementation by *Schlichtkrull et al. (2018)* as TensorFlow-RGCN (*TF-RGCN*). Our RGCN implementation is available at https://github.com/thiviyanT/torch-rgcn.

In this section, we will describe how we implemented the Relational Graph Convolutional Network. We begin by introducing crucial concepts for the implementation.

### Einstein summation

Message passing requires manipulating high-dimensional matrices and tensors using many different operations (*e.g.*, transposing, summing, matrix-matrix multiplication, tensor contraction). We will use Einstein summation to express these operations concisely.

Einstein summation (einsum) is a notational convention for simplifying tensor operations (*Kuptsov, 2022*). Einsum takes two arguments: (1) an equation[6] in which characters identifying tensor dimensions provide instructions on how the tensors should be transformed and reshaped, and (2) a set of tensors that need to be transformed or reshaped. For example, einsum ($ik, jk \rightarrow ij, A, B$) represents the following matrix operation:

$$C_{ij} = \sum_k A_{ik} \cdot B_{jk}, \tag{8}$$

The general rules of an einsum operation are that indices which are excluded from the result are summed out, and indices which are included in all terms are treated as the batch dimension. We use einsum operations in our implementation to simplify the message passing operations.

### Sparsity

Since many graphs are sparsely connected, their adjacency matrices can be efficiently stored on memory as *sparse tensors*. Sparse tensors are memory efficient because, unlike dense tensors, sparse tensors only store non-zero values and their indices. We make use of sparse matrix multiplications[7]. For sparse matrix operations on GPUs, the only multiplication operation that is commonly available is multiplication of a sparse matrix $S$ by a dense matrix $D$, resulting in a dense matrix. We will refer to this operation as spmm ($S, D$). For our implementation, we endeavour to express the sparse part of the RGCN message passing operation (Eq. (4)), including the sum over relations, in a single sparse matrix multiplication.

[5] TensorFlow 2 is not backward compatible with TensorFlow 1 code.

[6] We use a notation that maps directly to the way einstein summation is used in code, rather than the standard notation.

[7] Recent advances in CUDA implementations have made it possible to perform computations involving sparse matrix multiplications to run on the GPU.

## Stacking trick

Using nested loops to iteratively pass messages between all neighboring nodes in a large graph would be very inefficient. Instead, we use a trick to efficiently perform message passing for all relations in parallel.

Edge connectivity in a relational graph is represented as a three-dimensional adjacency tensor $A \in \mathbb{R}^{R^+ \times N \times N}$, where $N$ represents the number of nodes and $R^+$ represents the number of relations. Typically, message passing is performed using batch matrix multiplications as shown in Eq. (4). However, at the time of writing, batch matrix operations for sparse tensors are not available in most Deep Learning libraries. Using spmm is the only efficient operation available, so we *stack* adjacency matrices and implement the whole RGCN in terms of this operation.

We augment $A$ by stacking the adjacency matrices corresponding to the different relations $A_r$ vertically and horizontally into $A_v \in \mathbb{R}^{(N+R^+) \times N}$ and $A_h \in \mathbb{R}^{N \times (N+R^+)}$, respectively.

$$A_v = \begin{bmatrix} A_1 \\ A_2 \\ \vdots \\ A_{1'} \\ A_{2'} \\ \vdots \\ A_s \end{bmatrix} \tag{9}$$

$$A_h = \begin{bmatrix} A_1 & A_2 & \cdots & A_{1'} & A_{2'} & \cdots & A_s \end{bmatrix} \tag{10}$$

Here, $[\cdot]$ represents a concatenation operation, and $A_v$ and $A_h$ are both sparse matrices. By stacking $A_r$ either horizontally or vertically, we can perform message passing using sparse matrix multiplications rather than expensive dense tensor multiplications. Thus, this trick helps to keep the memory usage low.

Algorithm 1 shows how message passing is performed using a series of matrix operations. All these are implementations of the same operation, but with different complexities depending on the shape of the input. In Theorem 1 (see the Appendix), we prove that the the stacked formulation is equivalent to the original RGCN formulation.

1. If the inputs $X$ to the RGCN layer are one-hot vectors, X can be removed from the multiplication. The **featureless message passing** simply multiplies $A$ with $W$, because the node feature matrix $X$ is not given. Note that $X$, in this case, can also be modelled using an identity matrix $I$. However, since $AW = AIW$, we skip this step to reduce computational overhead.

2. In the **horizontal stacking approach**, $X$ multiplied with $W$. This yields the $XW$ tensor, which is then reshaped into a $NR^+ \times N$ matrix. The reshaped $XW$ matrix is then multiplied with $A_h$ using spmm.

3. In the **vertical stacking approach**, the $X$ is mixed with $A_v$ using spmm. The product is reshaped into a tensor of dimension $R^+ \times N \times N$. The tensor $AX$ is then multiplied with $W$.

---

**Algorithm 1: Message passing layer.**

**Input:** $A$, $\sigma$, $[X]$

**Result:** $H$

**if** *featureless* **then**
$\quad x \Leftarrow einsum(ni, io \rightarrow no, Ah, W)$
$\quad H \Leftarrow \sigma(x)$
**else**
$\quad$ **if** *horizontally stacked* **then**
$\quad\quad x \Leftarrow einsum(ni, rio \rightarrow rno, X, W)$
$\quad\quad$ reshape $x$ into a matrix with the dimensions $NR^+ \times N_{out}$
$\quad\quad x \Leftarrow spmm(A_h, x)$
$\quad$ **else**
$\quad\quad x \Leftarrow spmm(A_v, X)$
$\quad\quad$ reshape $x$ into a tensor with the dimensions $R^+ \times N \times N_{in}$
$\quad\quad x \Leftarrow einsum(rio, rni \rightarrow no, W, x)$
$\quad$ **end**
$\quad H \Leftarrow \sigma(x)$
**end**

---

Any dense/dense tensor operations is implemented with the einsum operation. For reasons highlighted in "Sparsity" it is desirable to construct $A_r$ as a sparse tensor. However, in PyTorch (*Paszke et al., 2019*) sparse/dense operations only allow multiplication of sparse matrix by dense matrix. We accept this as a limitation of the current version of PyTorch. To circumvent this issue, we use the stacking trick to construct $A_r$ as a sparse matrix and use the spmm (S, D) operation. Thus, the stacking trick is a memory efficient operation for the message passing operation in the RGCN layers.

Any dense/dense tensor operations can be implemented with einsum, but in the current version of PyTorch sparse/dense operations only allow multiplication of sparse matrix by dense matrix. Therefore, the stacking trick provides a memory efficient way to perform message passing operation in the RGCN layers (see Sparsity and Stacking trick). The vertical stacking approach is suitable for low dimensional input and high dimensional output, because the projection to low dimensions is done first. While the horizontal stacking approach is good for high dimensional input and low dimensional output as the projection to high dimension is done last. These matrix operations are visually illustrated in Fig. 4.

Thus far, we focused on how Relational Graph Convolutional layers work and how to implement them. As mentioned in "Literature review", RGCNs can be used for many downstream tasks. Now, we will discuss how these graph convolutional layers can be used as building blocks in larger neural networks to solve two downstream tasks implemented in the original RGCN article (*Schlichtkrull et al., 2018*): *node classification* and *link*

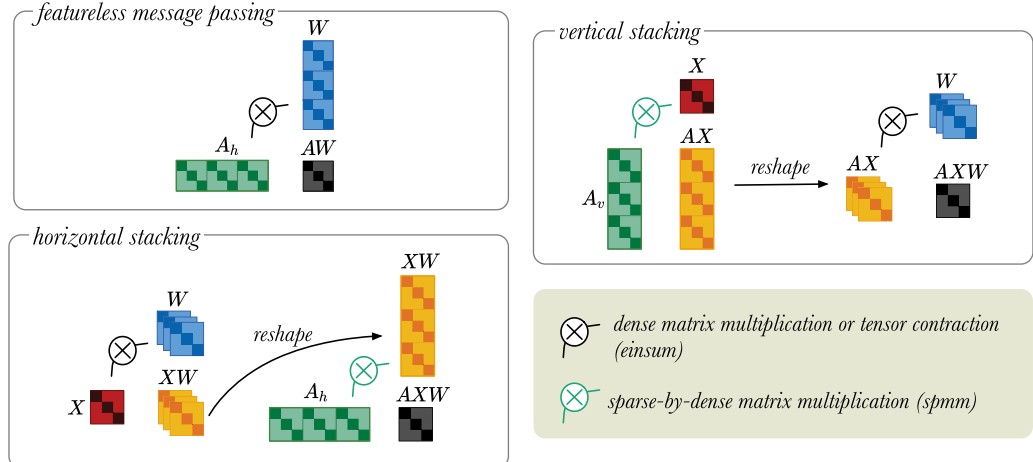

**Figure 4 A simplified visual representation of different message passing approaches: featureless-message passing (top left), message passing using horizontally stacked adjacency matrices (bottom left), and message passing using vertically stacked adjacency matrices (top right).** The black ⊗ indicates multiplication between dense tensors, which can be implemented with an einsum operator. The green ⊗ refers to sparse-by-dense multiplication, for which the spmm operation is required. Black arrow indicates tensor reshaping.

*prediction.* In the next two sections, we detail the model setup, our reproduction experiments and new configurations of the models. We begin with node classification.

# DOWNSTREAM TASK: NODE CLASSIFICATION

In the node classification task, the model is trained under a transductive setting which means that the whole graph, including the nodes in the test set, must be available during training, with only the labels in the test set withheld. The majority of the nodes in the graph are unlabelled and the rest of the nodes are labelled (we call these *target nodes*). The goal is to infer the missing class information, for example, that `Amsterdam` belongs to the class `City`.

## Model setup

Figure 5 shows the node classification models described in (*Schlichtkrull et al., 2018*), using a two-layer model. Full-batch training is used for training the node classification model, meaning that the whole graph is represented in the input adjacency matrix $A$ for the RGCN. The input is the unlabeled graph, the output are the class predictions and the true predictions are used to train the model. The first layer of the RGCN is ReLU activated and it embeds the relational graph to produce low-dimensional node embeddings. The second RGCN layer further mixes the node embeddings. Using softmax activation the second layer generates a matrix with the class probabilities, $Y \in \mathbb{R}^{N \times C}$, and the most probable classes are selected for each unlabelled node in the graph. The model is trained by optimizing the categorical cross entropy loss:

$$\mathcal{L} = -\sum_{i=1}^{y} \sum_{c=1}^{C} t_{ic} \ln \mathbf{h}_{ic}^{(L)}, \tag{11}$$

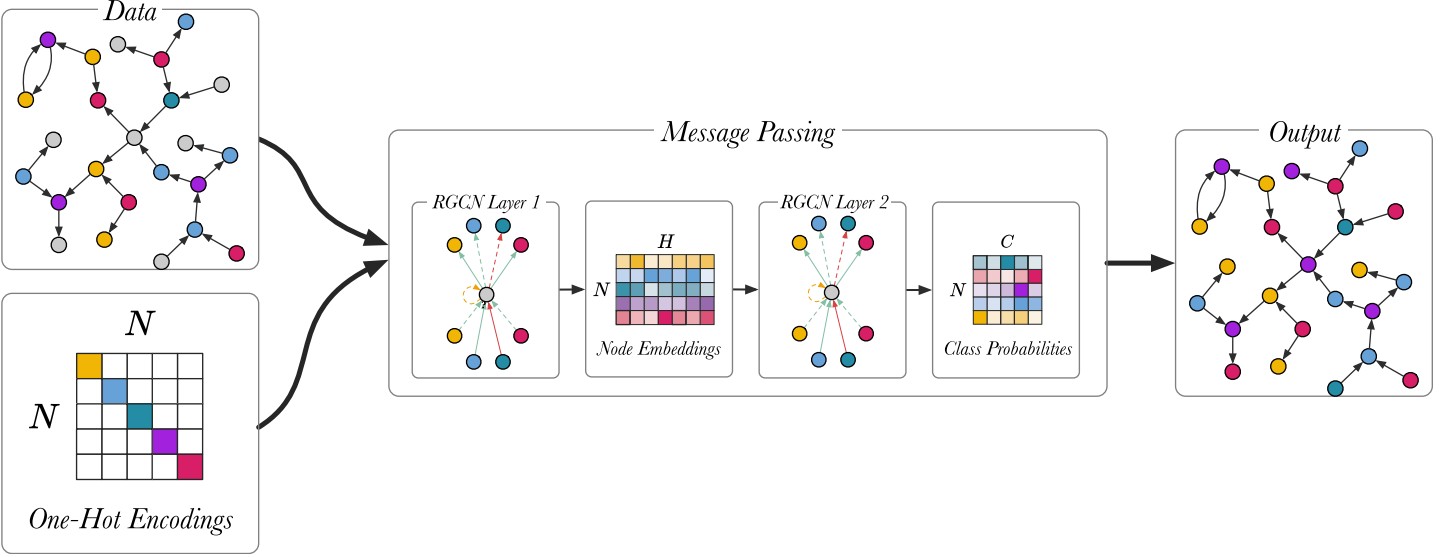

**Figure 5 An overview of the node classification model with a two-layer RGCN.** Different colour (magenta, green, blue yellow, violet) is used to highlight different entity types. Unlabelled entities are in grey.

where $\mathcal{Y}$ is the set of labelled nodes, $C$ represents the number of classes, $t_{ic}$ is one-hot encoded ground truth labels and $\mathbf{h}_{ic}^{(L)}$ represents node representations from the RGCN. The last layer ($L$) of the RGCN is softmax-activated. The trained model can be used to infer the classes of unlabelled nodes.

### e-RGCN

In GCNs (*Kipf & Welling, 2017*), the node features $X$ are represented by a matrix $X \in \mathbb{R}^{N \times F}$, where $N$ is the number of nodes and $F$ is the number of node features. When node features are not available, one-hot vectors can be used instead. An alternative approach would be to represent the features with continuous values $E \in \mathbb{R}^{N \times D}$, where $D$ is the node embedding dimension.

In the GCN setting (*Kipf & Welling, 2017*), using one hot vectors is functionally very similar to using embedding vectors: the multiplication of the one hot vector by the first weight matrix $W$, essentially selects a row of $W$, which then functions as an embedding of that node. In the RGCN setting, the same holds, but we have a separate weight matrix for each relation, so using one-hot vectors is similar to defining a separate node embedding for each relation. When we feed the RGCN a single node embedding for each node instead, we should increase the embedding dimension $D$ to compensate.

Initial experiments showed that simply replacing the node features $X$ with node embeddings $E$[8] results in a drop in performance on the node classification benchmark data. We believe that the additional parameters from $E$ gave the model more flexibility to overfit to training data and thus leading to an overall performance drop on the test dataset. After some experimentation, we ended up with the following model, which we call the *embedding-RGCN* (e-RGCN). Its message passing rule is described in Eq. (12). The weight

[8] Message passing rule:
$H = \sigma(\sum_{r=1}^{R} A_r E W_r)$.

[9] This is a special case of the block decomposition with $1 \times 1$ blocks.

matrix is restricted to a diagonal matrix (with all off diagonal elements fixed to zero)[9] and then the product is multiplied by the adjacency matrix.

$$h = \sigma\left(\sum_{r=1}^{R} A_r \, E \, \text{diag}(\mathbf{w}_r)\right), \tag{12}$$

Here, $E$ is the node embeddings broadcasted across all the relations $\mathcal{R}$, $w_r$ is a vector containing weight parameters for relation $r$. Here, diag($\cdot$) is a function that takes a vector $\mathbf{l} \in \mathbb{R}^Q$ as an input and outputs a diagonal matrix $N \in \mathbb{R}^{Q \times Q}$, where the diagonal elements are elements from the original vector $L$.

Using a diagonal weight matrix improves parameter efficiency, while enabling distinction between relations. We created a new node classification model, where the first layer is an e-RGCN layer and the second layer is a standard RGCN (without regularisation) that predicts class probabilities. This model provides competitive performance with the RGCN, using only 8% of the parameters.

## Node classification experiments
### Datasets

We reproduce the node classification experiments using the benchmark datasets that were used in the original article: AIFB (*Bloehdorn & Sure, 2007*), MUTAG (*Debnath et al., 1991*), BGS (*de Vries, 2013*) and AM (*de Boer et al., 2012*). We also evaluate e-RGCN on the same datasets. AIFB is a dataset that describes a research institute in terms of its staff, research group, and publications. AM (Amsterdam Museum) is a dataset containing information about artifacts in the museum. MUTAG is derived as an example dataset for the machine learning model toolkit about complex molecules. The BGS (British Geological Survey) dataset contains information about geological measurements in Great Britain.

Since the messages in a two-layer RGCN cannot propagate further than two hops, we can prune away the unused nodes from the graph. This significantly reduces the memory usage for large datasets (BGS & AM) without any performance deterioration. In Theorem 2 (see Appendix), we mathematically show that any nodes that are more than $k$ hops away from the target nodes does not affect the outcome of the message passing. To the best of our knowledge, this was first implemented in the DGL library (*Wang et al., 2020*). For the AM and BGS datasets, the graph was pruned by removing any nodes that are two hops away from the target nodes. Pruning significantly reduces the number of entities, relations and edges and thus, lowers the memory consumption of the node classification model, making it feasible to train it on a GPU with 12 GB of memory. Table 1 shows the statistics for the node classification datasets. We use the same training, validation and test split as in (*Schlichtkrull et al., 2018*).

### Details

All node classification models were trained following (*Schlichtkrull et al., 2018*) using full-batch gradient descent for 50 epochs. However, we used 100 epochs for e-RGCN on the AM dataset. Glorot uniform initialisation (*Glorot & Bengio, 2010*) was used to initialise parameters with a gain of $\sqrt{2}$ corresponding to the ReLU activation function. Kaiming

**Table 1 Statistics about the node classification benchmark datasets.**

| Dataset | AIFB | MUTAG | BGS* | AM* |
|---|---|---|---|---|
| Entities | 8,285 | 23,644 | 87,688 | 246,728 |
| Relations | 45 | 23 | 70 | 122 |
| Edges | 29,043 | 74,227 | 230,698 | 875,946 |
| Labeled | 176 | 340 | 146 | 1,000 |
| Classes | 4 | 2 | 2 | 11 |

Notes:
Number of entities, relations, edges and classes along with the number of labeled entities for each of the datasets. *Labeled* denotes the subset of *entities* that have labels and entities are the nodes without any labels.
* Entities more than two hops away from the target label were pruned.

**Table 2 Node classification accuracy for TF-RGCN, Torch-RGCN and e-RGCN.**

| Dataset | Model accuracy (%) | | |
|---|---|---|---|
| | TF-RGCN | Torch-RGCN | e-RGCN |
| AIFB | 95.83 ± 0.62 | 95.56 ± 0.61 | 89.17 ± 0.28 |
| AM | 89.29 ± 0.35 | 89.19 ± 0.35 | 89.04 ± 0.25 |
| BGS | 83.10 ± 0.80 | 82.76 ± 0.89 | 81.72 ± 0.53 |
| MUTAG | 73.23 ± 0.48 | 73.38 ± 0.60 | 71.03 ± 0.58 |

Note:
Results for TF-RGCN were taken from the original article. These are averages over 10 runs, with standard deviations.

initialization (*He et al., 2015*) was used to initialise the node embeddings in the e-RGCN node classification model[10]. Basis decomposition was used for the RGCN-based node classification[11]. All RGCN and e-RGCN models, except for the e-RGCN on the AM dataset, were trained using a GPU.

### Results

Table 2 shows the results of the node classification experiments in comparison to the original RGCN article. Torch-RGCN achieves similar performances to TF-RGCN reported in *Schlichtkrull et al. (2018)*. We observed that the training times[12] of the node classification models largely depended on the size of the graph dataset. The CPU training times varied from 45 s for the AIFB dataset to 20 min for the AM dataset. Since our implementation makes use of GPU's, we were able to run the Torch-RGCN models on a GPU and train the model within a few minutes.

## DOWNSTREAM TASK: LINK PREDICTION

We now turn towards the second task performed in the original article, multi-relational link prediction. The aim is to learn a scoring function that assigns true triples high scores and false triples low scores (*Bordes et al., 2013*), with the correct triple ranking the highest. After training, the model can be used to predict which missing triples might be true, or which triples in the graph are likely to be incorrect.

[10] The gain parameter is taken from the DGL implementation (*Wang et al., 2020*). The original implementation does not appear to apply a gain. This choice does not seem to affect the classification performance.

[11] 40 bases for the AM & BGS. 30 bases for MUTAG.

[12] We measured the wall time of the Python scripts from start to finish, which includes training and evaluation of the models.

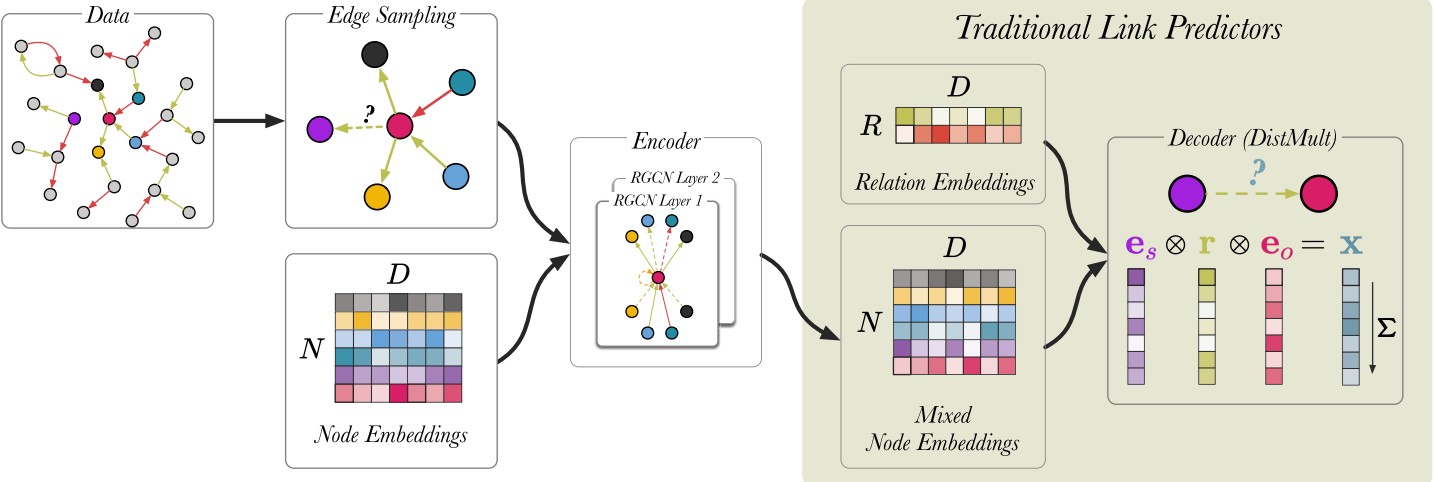

**Figure 6** **A schematic visualisation of link prediction models.** Edges are coloured (red and green) to indicate different edge labels. RGCN-based encoders can be seen as an extension to traditional link predictors, such as DistMult (*Yang et al., 2015*) and TransE (*Bordes et al., 2013*). RGCNs enrich the node representations used by these models by mixing them along the edges of the graph, before applying the score function. In this case, removing the RGCN layers and the upstream edge sampling, recovers the original DistMult. In the last step, the vectors corresponding to entities and relation are element-wise multiplied and the product $x$ is summed. For a given triple $\langle s, r, o \rangle$, the model produces a single scalar value $x$ which indicates how likely the triple is to be true.

## Model setup

We follow the procedure outlined by *Schlichtkrull et al. (2018)*. Figure 6 shows a schematic representation of the link prediction model as described in the original article. During training, traditional link predictors (*Bordes et al., 2013*; *Yang et al., 2015*) simultaneously update node representations and learn a scoring function (decoder) that predicts the likelihood of the correct triple being true. RGCN-based link predictors introduce additional steps upstream.

We begin by sampling 30,000 edges from the graph using an approach called *neighborhood edge sampling* (see Edge sampling). Then, for each triple we generate 10 negative training examples, generating a batch size of 330,000 edges in total. Node embeddings $E \in \mathbb{R}^{N \times D}$ are generated using a standard embedding layer and are used as an input for the RGCN[13]. The RGCN performs message passing over the sampled edges and generates mixed node embeddings. Finally, the DistMult scoring function (*Yang et al., 2015*) uses the mixed node embeddings and relation embeddings to compute the likelihood of a link existing between a pair of nodes. For a given triple $\langle s, r, o \rangle$, the model is trained by scoring all potential combination of the triple using the function:

$$f(s, r, o) = \sum_i \left( \mathbf{e}_s \mathbf{r} \mathbf{e}_o \right)_i = x \tag{13}$$

Here, $\mathbf{e}_s$ and $\mathbf{e}_o$ are the corresponding node embedding of entities $s$ and $o$, generated by the RGCN encoder. $\mathbf{r}$ is a low-dimensional vector of relation $r$, which is generated by the DistMult decoder. Relation embeddings $R \in \mathbb{R}^{R^+ \times D}$ are generated using a standard

[13] In the original implementation, the embeddings are implemented as affine operation (*i.e.*, biases are included) and they are ReLU activated. We reproduce this behaviour but it is not clear whether this gives any benefits over simple, unactivated embeddings (as used in the e-rgcn).

embedding layer. As *Schlichtkrull et al. (2018)* highlighted in their work, the DistMult decoder can be replaced by any Knowledge Graph scoring function.

Similar to previous work on link prediction (*Yang et al., 2015*), the model is trained using negative training examples. For each observed (positive) triple in the training set, we randomly sample 10 negative triples (*i.e.*, we use a negative sampling rate of 10). These samples are produced by randomly corrupting either the subject or the object of the positive example (the probability of corrupting the subject is 50%). Binary cross entropy loss[14] is used as the optimization objective to push the model to score observable triples higher than the negative ones:

$$\mathcal{L} = - \sum_{(s,r,o,y) \in \mathcal{T}} y \log l(f(s,r,o)) + (1-y) \log(1 - l(f(s,r,o))), \quad (14)$$

where $\mathcal{T}$ is the total set of positive and negative triples, $l$ is the logistic sigmoid function, and $y$ is an indicator set to $y = 1$ for positive triples and $y = 0$ for negative triples. $f(s, r, o)$ includes entity embeddings from the RGCN encoder and relations embeddings from the DistMult decoder.

### Edge dropout

In their work, *Schlichtkrull et al. (2018)* apply edge dropout to the link prediction model which acts as an additional regularisation method. This involves randomly selecting edges and removing them from a graph. As described in "Relational graph convolutional network", for every edge in the graph inverse edges $A_{r'}$ and self-loops $A_s$ are added within the RGCN layer. Dropping edges after this step poses a potential data leakage issue because inverse edges and self-loops of dropped edges will be included in the message passing step and thus, invalidate the model performance. To circumvent this issue, edges are dropped from a graph before feeding it into the RGCN.

Edge dropout is applied such that the dropout rates on the self-loops $\mathcal{R}_s$ are lower than for the data edges $\mathcal{R}$ and inverse edges $\mathcal{R}'$. One way to think about this that this ensures that the message from a node to itself is prioritised over incoming messages from neighboring nodes. In our implementation, we separate out $A_s$ from $A$ and then apply the different edge dropout rates separately. The edge dropout is performed before row-wise normalising $A$.

### Edge sampling

Graph batching is required for training the RGCN-based link prediction model, because it is computationally expensive to perform message passing over the entire graph due to the large number of hidden units used for the RGCN[15].

*Schlichtkrull et al. (2018)* sample an edge with the probability proportional to its weight. In *uniform edge sampling*, equal weights are given to all the edges. However, in *neighborhood edge sampling*, initial weights are proportional to the node degrees of vertices connected to edges. Then as edges are being sampled, the weight of its neighboring edges is increased and this increases the probability of these edges being sampled (*Klusowski & Wu, 2018*). This sampling approach benefits link prediction because the neighboring edges provide context information to deduce the existence of a relation between a given pair of

[14] *Schlichtkrull et al. (2018)* multiply their loss by $\frac{1}{(1+\omega)|\hat{\varepsilon}|}$. $\omega$ is the negative sampling rate and $|\hat{\varepsilon}|$ is the number of edges sampled. We leave this term out of our implementation, because it is a constant and thus it would not affect the training.

[15] *Schlichtkrull et al. (2018)* use a large number of hidden units: 200 for FB15k and WN18; 500 for FB15k-237. In the node classification fewer hidden units are required in the RGCN. This makes the link prediction model more memory demanding.

**Table 3 Statistics about link prediction benchmark datasets.**

| Dataset | WN18 | FB-Toy |
|---|---|---|
| Entities | 40,943 | 280 |
| Relations | 18 | 112 |
| Train edges | 141,442 | 4,565 |
| Val. edges | 5,000 | 109 |
| Test edges | 5,000 | 152 |

Note:
 Number of entities and relation types along with the number of edges per split for the three datasets.

entities. In contrast, uniform edge sampling assumes that all edges are independent of each other, which is not applicable to Knowledge Graphs.

## Link prediction experiments

As in the original article, the models are evaluated using Mean Reciprocal Rank (MRR) and Hits@$k$ ($k$ = 1, 3 or 10). The Torch-RGCN model was trained for 7,000 epochs without early stopping[16]. We monitored the training by evaluating the model at regular intervals (every 500 epochs) using the validation dataset. Schlichtkrull initialisation (*see Appendix*) was used to initialise all parameters in the link prediction models and in our reproductions. *Schlichtkrull et al. (2018)* trained their models on the CPU. Our Torch-RGCN implementation can be trained using GPU acceleration. Early stopping was not used. We used the hyperparameters described in (*Schlichtkrull et al., 2018*).

### Datasets

To evaluate link prediction, *Schlichtkrull et al. (2018)* used subsets of Freebase (FB-15k and FB15k-237), and WordNet (WN18) (*Bordes et al., 2013*). We only use WN18[17]. WN18 is a subset of WordNet, a graph which describes the lexical relations between words. To check our reproduction, we also used FB-Toy (*Ruffinelli, Broscheit & Gemulla, 2020*) which was not in the original article. FB-Toy is a dataset consisting of a subset of FB15k. In Table 3, we show the statistics corresponding to these graph datasets.

### Details

For link prediction, a single-layer RGCN with basis decomposition for WN18 and for FB-Toy a two-layer RGCN with block diagonal decomposition is used. An $\mathcal{L}2$ regularisation penalty of 0.01 for the scoring function is applied. To compute the *filtered* link prediction scores, triples that occur in the training, validation and test are filtered. The Torch-RGCN model is trained on the WN18 and FB-Toy datasets using batched training. Edges are randomly sampled from a Knowledge Graph using the neighborhood edge approach. A total of 30,000 neighboring edges are randomly sampled at every epoch for the WN18 dataset (*Schlichtkrull et al., 2018*), and for FB-toy dataset, we sampled 300 neighboring edges[18]. An edge dropout rate of 0.2 is applied for self-loops and 0.5 for data edges and inverse edges. In our reproduction attempts, we have found that this approach enables the model to perform better than uniform edge sampling. The RGCN is initialised using

[16] Personal communication with the author confirmed that early stopping was not used in the original work.

[17] The link prediction model was expensive to train (3–5 days of training for a single model). WN18 was sufficient to establish reproduction.

[18] We sampled only 300 edges because the FB-toy dataset was roughly 100 times smaller than the WN18 dataset (in terms of number of nodes).

Schlichtkrull normal initialisation (*see Appendix*), while the DistMult scoring function is initialised using standard normal initialisation.

We follow the standard protocol for link prediction. See *Ruffinelli, Broscheit & Gemulla (2020)* for more details. Some of the hyperparameters used in training were not detailed in *Schlichtkrull et al. (2018)*. To the furthest extent possible, we followed the same training regime as the original article and code base, and we recovered missing hyperparameters. The hyperparameters for all experiments are provided in documented configuration files on https://github.com/thiviyanT/torch-rgcn.

### Results

We verify the *correctness* of our implementation by reproducing the performance on a small dataset (FB-Toy) and by comparing the statistics of various intermediate tensors in the implementation with those of the reference implementation (https://github.com/MichSchli/RelationPrediction). We selected a number of intermediate tensors in the link prediction model in our implementation and the original implementation. Then, we measured the statistics of the intermediate tensors. In Table 4 we report the statistics of the intermediate tensors for TF-RGCN (original model) and Torch-RGCN (our implementation) link prediction models. These results suggests that the parameters used by both models came from a similar distribution and thus verified that they are one-to-one replication.

After confirming that our reproduction is correct, we attempted to replicate the link prediction results on the WN18 dataset[19]. Table 5 also shows the results of the link prediction experiments carried out. Our Torch-RGCN implementation scored lower than the TF-RGCN model and therefore, we were unable to duplicate the exact results reported in the original article. The exact hyperparameters that *Schlichtkrull et al. (2018)* used in their experiments were not publicly available. We believe that the discrepancies between the Torch-RGCN scores and the TF-RGCN scores was caused as a result of using different hyperparameter configurations.

Despite our best efforts, we were unable to reproduce the exact link prediction results reported in the original article (*Schlichtkrull et al., 2018*). This is due to the multitude of hyperparameters[20], not all of which are specified in the article, and the long time required to train the model, with runtimes of several days. We did however manage to show the correctness of our implementation using a small-scale experiment. We consider this an acceptable limitation of our reproduction, because the current training time of the RGCN compared to the state of the art KGE models (*Ruffinelli, Broscheit & Gemulla, 2020*). A Distmult embedding model can be trained in well under an hour on any of the standard benchmarks, and as shown by *Ruffinelli, Broscheit & Gemulla (2020)*, outperforms the RGCN by a considerable margin. Thus, the precise link prediction architecture described in *Schlichtkrull et al. (2018)* is less relevant in the research landscape.

### c-RGCN

The link prediction architecture presented in *Schlichtkrull et al. (2018)* does not represent a realistic competitor for the state of the art and is very costly to use on large graphs.

[19] Runs on FB15k and FB15k-237 took from 3 to 5 days to complete training.

[20] There are at least 10 non-trivial hyperparameters: Number of Epochs, Learning Rate, Graph Batch Size, Negative Sampling Rate, Number of RGCN layers, Dimension of RGCN layers, Weight Decomposition Method, Number of Blocks or number of basis functions, Edge Dropout Rate, $\mathcal{L}2$ regularisation penalty for the scoring function.

**Table 4 Parameter statistics for intermediate products at various points in link prediction model for the FB-Toy dataset.**

| No. | Intermediate tensors | Tensor dimensions | | Parameter statistics | |
|---|---|---|---|---|---|
| | | | | TF-RGCN | Torch-RGCN |
| 1 | Node Embeddings: Initialisation | $280 \times 500$ | min | −0.45209 | −0.48939 |
| | | | max | 0.45702 | 0.48641 |
| | | | mean | −0.00004 | 0.00029 |
| | | | std | 0.10697 | 0.10765 |
| 2 | Node Embeddings: Output | $280 \times 500$ | min | 0.0 | 0.0 |
| | | | max | 0.47124 | 0.51221 |
| | | | mean | 0.04318 | 0.04290 |
| | | | std | 0.06301 | 0.06273 |
| 3 | RGCN Layer 1: Block Initialisation for data edges and inverse edges[†] | $224 \times 100 \times 5 \times 5$ | min | −1.38892 & −1.20551 | −1.28504 |
| | | | max | 1.47686 & 1.3872786 | 1.26404 |
| | | | mean | 0.00007 & −0.00008 | 0.00001 |
| | | | std | 0.27726 & 0.27692 | 0.27715 |
| 4 | RGCN Layer 1: Block Initialisation for self-loops | $500 \times 500$ | min | −1.23380 | −1.20324 |
| | | | max | 1.30949 | 1.16375 |
| | | | mean | −0.00049 | −0.00095 |
| | | | std | 0.27716 | 0.27755 |
| 5 | RGCN Layer 1: Output | $280 \times 500$ | min | −2.58617 | −2.75152 |
| | | | max | 2.43774 | 2.63124 |
| | | | mean | 0.02317 | 0.00799 |
| | | | std | 0.51760 | 0.53759 |
| 6 | DistMult: Relation Initialisation | $112 \times 500$ | min | −4.12359 | −3.97444 |
| | | | max | 4.89700 | 3.95794 |
| | | | mean | −0.00947 | −0.00186 |
| | | | std | 0.99675 | 0.99851 |
| 7 | DistMult: Output | $3,300 \times 1$ | min | −27.21030 | −30.75097 |
| | | | max | 27.0885849 | 25.89389 |
| | | | mean | 0.03524 | 0.78507 |
| | | | std | 7.75823 | 7.27595 |

**Notes:**
We report the minimum, maximum, mean and the standard deviation of the distributions.
[†] *Schlichtkrull et al. (2018)* used separate weight matrices for data edges $\mathcal{R}$ and inverse edges $\mathcal{R}'$, resulting in two intermediate tensors with the dimensions $112 \times 100 \times 5 \times 5$. Thus, we report the statistics for these two tensors.

**Table 5 Mean reciprocal rank (MRR) and Hits@k ($k$ = 1, 3 and 10) for link prediction using RGCN, Torch-RGCN and c-RGCN.**

| Dataset | Model | MRR | Hits@1 | Hits@3 | Hits@10 |
|---|---|---|---|---|---|
| WN18 | TF-RGCN | 0.814 | 0.686 | 0.928 | 0.955 |
| | Torch-RGCN | 0.749 | 0.590 | 0.908 | 0.939 |
| | c-RGCN | 0.717 | 0.558 | 0.867 | 0.933 |

**Note:**
Triples from the truth set (train, validation and test set) have been filtered. Results for TF-RGCN was taken from the original article.

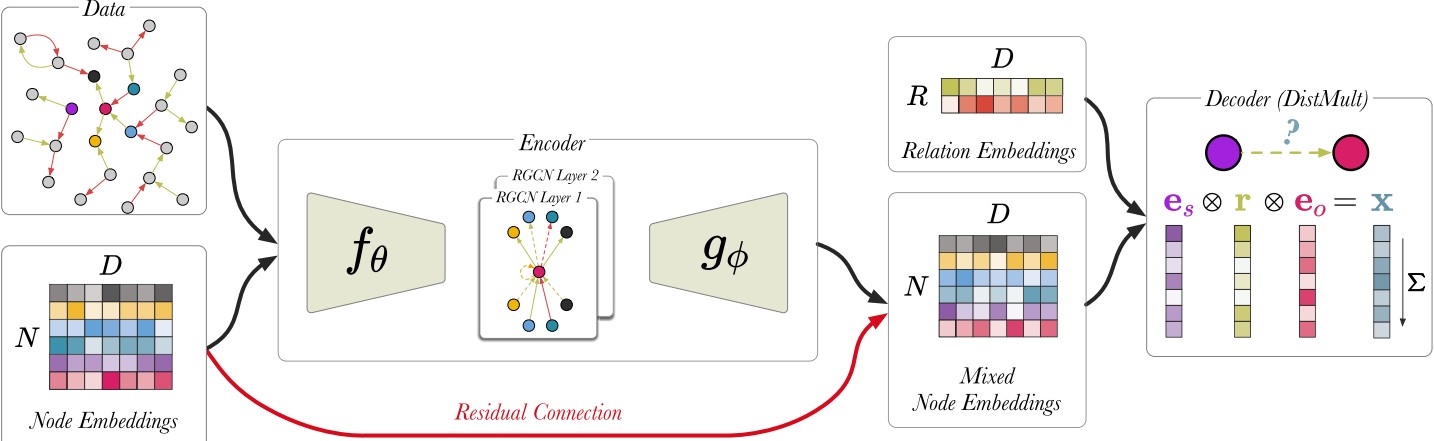

**Figure 7** **A schematic visualisation of c-RGCN based link prediction model.** Here, the encoder has a bottleneck architecture. $f_\theta$ and $g_\phi$ are linear layers. Prior to message passing $f_\theta$ compresses the input node embeddings, and then $g_\phi$ projects the mixed node embeddings back up to their original dimensions. The red arrow indicates the residual connection. All edges from the training set are used (*i.e.*, edge sampling is not required).

Furthermore, a problem with the original RGCN link predictor is that we need high dimensional node representations to be competitive with traditional link predictors, such as DistMult (*Yang et al., 2015*), but the RGCN is expensive for high dimensions. However, we do believe that the basic idea of message passing is worth exploring further in a link prediction setting.

To show that there is promise in this direction, we offer a simplified link prediction architecture that uses a fraction of the parameters of the original implementation (*Schlichtkrull et al., 2018*) uses. This variant places a bottleneck architecture around the RGCN in the link prediction model, such that the node embedding matrix $E$ is projected down to a lower dimension, $C$, and then the RGCN performs message propagation using the compressed node embeddings. Finally, the output is projected up back to the original dimension, $D$, and computes the DistMult score from the resulting high-dimensional node representations. We call this encoding network the *compression-RGCN* (c-RGCN).

Figure 7 shows a schematic representation of the c-RGCN model. Equations (15) and (16) show the message passing rule for the first and second layer of the c-RGCN encoder, respectively. We selected a node embedding size of 128 and compressed it to a vector dimension of 16. We also include a residual connection by including $E$ in the second layer of the c-RGCN. The residual connection allows the model, in principle, to revert back to DistMult if the message passing adds no value. If all RGCN weights are set to 0, we recover the original DistMult. The c-RGCN model can be trained on the GPU.

$$H^1 = \sigma\left(\sum_{r=1}^{R^+} A\, W_r\, f_\theta(E)\right), \tag{15}$$

where $f(X) = XW_\theta + b$ with $W_\theta \in \mathbb{R}^{C \times D}$.

$$H^2 = E + g_\phi\left(\sigma\left(\sum_{r=1}^{R^+} A \; W_r \; H^1\right)\right), \tag{16}$$

where $g(X) = XW_\phi + b$ with $W_\phi \in \mathbb{R}^{D \times C}$.

As shown in Table 5, the c-RGCN does not perform much worse than the original implementation. However, it is much faster, and memory efficient enough for full batch evaluation on the GPU. There is a clear trade-off between compression size of the node embeddings and the performance in link prediction. While this result is far from a state of the art model, it serves as a proof-of-concept that there may be ways to configure RGCN models for a better performance/efficiency tradeoff.

## DISCUSSION

We now discuss the implications for the use of the RGCN model, the performance of the new variants and the lessons learned from this reproduction.

### Implications for RGCN usage

We believe that Relational Graph Convolutional Networks are still very relevant because it is one of the simplest members of the message passing models and is a good starting place for exploration of machine learning for Knowledge Graphs.

RGCNs clearly perform well on node classification tasks because the task of classifying nodes benefits from message passing. This means that a class for a particular node is selected by reasoning about the classes of neighboring nodes. For example, a researcher can be categorised into a research domain by reasoning about information regarding their research group and close collaborators. There is no direct way to perform node classification using traditional Knowledge Graph Embeddings (KGE) models, such as TransE and DistMult.

While the RGCN is a promising framework, in its current setting we found that the link prediction model proposed by *Schlichtkrull et al. (2018)* is not competitive with current state of the art (*Ruffinelli, Broscheit & Gemulla, 2020*) and the model is too expensive with considerably lower performance. In our article, we clarify that RGCN-based link predictors are extensions of KGE models (*Ruffinelli, Broscheit & Gemulla, 2020*), thus training RGCN to predict links will always be more expensive than using a state of the art KGE model. RGCN-based link predictor take several days to train, while state of the art relation models run in well under an hour (*Ruffinelli, Broscheit & Gemulla, 2020*).

To aid the usage of RGCN, we presented two new configurations of the RGCN:

**e-RGCN.** We propose a new variant of the node classification model which uses significantly less parameters by exploiting a diagonal weight matrix. Our results indicate that e-RGCN has the potential to perform competitively with the original model and deserves further investigation. The potential advantage of e-RGCN is that it can operate on graphs that are much larger than the benchmark datasets we used in our reproduction studies.

**c-RGCN.** We also present a proof-of-concept model that performs message passing over compressed graph inputs and thus, improves the parameter efficiency for link

prediction. The c-RGCN has several advantages over the regular RGCN link predictor: (1) c-RGCN does not require sampling edges, because it is able to process the entire graph in full-batch, (2) c-RGCN takes a fraction of the time it takes to train an RGCN link predictor, (3) c-RGCN uses fewer parameters, and (4) it is straightforward to implement. Although the results for the c-RGCN are not as strong, this sets a path for further development towards efficient message models for relational graphs.

## Reproduction

Evolving technologies pose several challenges for the reproducibilty of research artifacts. This includes frequent updates being made to existing frameworks, such as PyTorch and TensorFlow, often breaking backward compatibility. We were in a strong position to execute this reproduction: (1) an author of this article also worked on the original article, (2) we contacted one of the lead authors of this article who was very responsive and (3) we were able to run the original source code (https://github.com/tkipf/relational-gcn and https://github.com/MichSchli/RelationPrediction) inside a virtual environment. Nevertheless, we found it considerably challenging to make a complete reproduction. To explain why and to contribute to avoiding such situations in the future, we briefly outline the lessons we have learned during the reproduction.

**Parameter Statistics.** There were discrepancies between the description of the link prediction model in *Schlichtkrull et al. (2018)* and the source code. The source code reproduces the values similar to the MRR scores reported in *Schlichtkrull et al. (2018)*. Thus, to reproduce the results we had to perform a deep investigation of the source code. Using the original source, we relied on comparing the parameter statistics and tensor sizes at various points in both models. Since these statistics are helpful to verify the correctness of an implementation, we believe this is a useful practice in aiding reproduction. For complex models with long runtimes, an overview of descriptive statistics of parameter and output tensors for the first forward pass can help to check implementation without running full experiments. We are publishing statistics for intermediate products that we obtained for the link prediction models (see Table 4).

**Small dataset.** We found that the link prediction datasets used by *Schlichtkrull et al. (2018)* were large and thus, impractical for debugging RGCN because it is costly to train them on large graphs. Using a smaller dataset (FB-Toy (*Ruffinelli, Broscheit & Gemulla, 2020*)) would enable quicker testing with less memory consumption. Thus, we report link prediction results on the FB-Toy dataset (see Table A2).

**Training times.** The training times were variable and strongly depended on the size of the graph, the number of relations and the number of epochs. *Schlichtkrull et al. (2018)* reported the computational complexity, but not practical training times. It turns out that this is an important source of uncertainty in verifying whether re-implementations are correct. We measured the runtimes, which includes training the model and using the pre-trained model for making inference. For 7,000 epochs, the link prediction runtimes for Torch-RGCN and c-RGCN on the WN18 dataset are 2,407 and 53 min, respectively. Node classification experiments took a few minutes to complete, because they only required 50–100 epochs. We encourage authors to report such concrete training times.

**Hyperparameter Search.** We found that hyperparameters reflect the complexity of the individual datasets. For example, AIFB, the smallest dataset, was not prone to overfitting. Whereas, the larger AM dateset required basis decomposition and needs a reduced hidden layer size. For link prediction, we were unable to identify the optimum hyperparameters for WN18, FB15k and FB15k-237 due to the sheer size of the hyperparameter space and long training times. We provide a detailed list of hyperparameter we use in our reproduction. While this is becoming more common in the literature, this serves as further evidence of the importance of this detailed hyperparameter reporting.

**Other factors.** We still faced the common challenges in software reproduction that others have long noted (*Fokkens et al., 2013*), including missing dependencies, outdated source code, and changing libraries. An additional challenge with machine learning models is that hardware (*e.g.*, GPUs) now also can impact the performance of the model itself. For instance, while we were able to run the original link prediction code in TensorFlow 1.4, the models no longer seemed to benefit from the available modern GPUs. Authors should be mindful that even if legacy code remains executable for a long time, executing it efficiently on modern hardware may stop being possible much sooner. Here too, reporting results on small-scale experiments can help to test reproductions without the benefit of hardware acceleration.

## CONCLUSION

We have presented a reproduction of the Relational Graph Convolutional Network and, using the reproduction, we provide a friendly explanation of how message passing works. Our new implementation of RGCN using PyTorch, TorchRGCN, is made openly available to the community. Furthermore, we also highlight subtleties of the RGCN that are crucial for its understanding, use and efficient implementation. While message passing is evidently useful for node classification, our findings also show that RGCN-based link predictors are currently too costly to make for a practical alternative to the state of the art. However, we believe that improving the parameter efficiency RGCNs could potentially make it more accessible. We present two novel configurations of the RGCN: (1) e-RGCN, which introduces node embeddings into the RGCN using fewer parameters than the original RGCN implementation, and (2) c-RGCN, a proof-of-concept model which compresses node embeddings and thus speeds up link prediction. These configurations provide the foundation for future work. We believe that the techniques proposed in this article may also be important for others implementing other message passing models. We hope that this can help serve the community in the use, development and research of this interesting model for machine learning on Knowledge Graphs.

## APPENDIX

### Proofs

**Definition 1.** *We define a relational graph $\mathcal{G}$ as a directed graph with labelled nodes and edges. F is the number of node features, R is the number of relations in the graph and N is the number of nodes in the graph. $A^r \in \mathbb{R}^{R \times N \times N}$ is a tensor that describes the edge connectivity*

for every relation. $A^h = \begin{bmatrix} A_1 \\ A_2 \\ \vdots \\ A_R \end{bmatrix}$ is a matrix where the adjacency matrices corresponding to

different relations are vertically stacked, and $A^v = [A_1 \ A_2 \ \cdots \ A_R]$ is a matrix where the adjacency matrices corresponding to different relations are horizontally stacked. $X \in \mathbb{R}^{N \times F}$ is a feature matrix. $W^r$ is a relation-specific weight matrix.

**Theorem 1.** *The vertically and horizontally stacked versions of the RGCN are equivalent to the original definition.*

*Proof.* To show that the stacking trick yields the same operation as the original RGCN definition.

First, we note that the non-linearity $\sigma$ is an element-wise operation. We can ignore this, and show that the input to the non-linearity is the same in all three cases.

To show this, we first express all three definitions in terms of the individual matrix elements. Writing the original R-GCN as

$$G = \sum_{r=1}^{R} A^r \cdot X \cdot W^r$$

where $X$ is the matrix of inputs, we can rewrite this operation in terms of the individual elements of $G$ as

$$G'_{r,i,j} = \sum_{k=1}^{N} A^r_{r,i,k} \cdot X_{r,k,j}$$

$$G''_{r,i,j} = \sum_{k=1}^{F} G'_{r,i,k} \cdot W_{r,k,j}$$

$$G_{r,i,j} = \sum_{k=1}^{F} G''_{r,i,j}.$$

Here, the first line shows a basic matrix multiplication, expressed element-wise $((AB)_{i,j} = \sum_{k} A_{i,k} \cdot B_{k,j})$, the second shows another matrix multiplication, and the third sums the result over all relations.

For the vertically stacked RGCN, we can write the computation of the output matrix $V$ as

$$V'_{i,j} = \sum_{k=1}^{N} A^v_{i,k} \cdot X_{k,j}$$

$$V''_{r,i,j} = V'_{(r-1)n+i,j}$$

$$V'''_{r,i,j} = \sum_{k=1}^{F} V''_{r,i,j} \cdot W_{r,k,j}$$

$$V_{i,j} = \sum_{r=1}^{R} V'''_{r,i,j}.$$

The first line again shows a matrix multiplication, but this time of the vertically stacked adjacency matrix with the feature matrix $X$. The second line reshapes this into a three-tensor separating our the relations. The third line is again a standard matrix multiplication and the final line sums out the relation.

We note that $V''_{r,i,j} = \sum_{k=1}^{F} A^v_{(r-1)n+i,k} X_{k,j} = \sum_{k=1}^{F} A^r_{r,i,k} X_{k,j} = G'_{i,j}$. It follows that $V''' = G''$ and $V = G$.

For the horizontally stacked RGCN, we can write the computation of the output matrix $H$ as

$$H'_{r,i,j} = \sum_{k=1}^{F} X_{r,i,k} \cdot W_{r,k,j}$$
$$H''_{(r-1)n+i,j} = H'_{r,i,j}$$
$$H_{i,j} = \sum_{k=1}^{RN} A^h_{i,k} \cdot H''_{k,j}.$$

The first line shows a matrix multiplication of the input features by the weight matrix, per relation. The second line shows the reshaping of the result into a vertical stack of matrices and the final line shows a single matrix multiplication between the horizontally stacked adjacency matrix and this vertical stack.

We complete the proof by noting that

$$H_{i,j} = \sum_k A^h_{i,k} \cdot H''_{k,j} = \sum_k \sum_r A^r_{i,k} \cdot H'_{r,k,j}$$
$$= \sum_k \sum_r A^r_{i,k} \sum_m X_{r,k,m} \cdot W_{r,m,j} = \sum_r \sum_m W_{r,m,j} \sum_k A^r_{i,k} X_{r,k,m}$$
$$= \sum_r \sum_m W_{r,m,j} G'_{r,i,m} = \sum_r G''_{r,i,j} = G_{i,j} \quad.$$

For the following theorem we define a relational graph $\mathcal{G}$ as a directed graph with labelled nodes and edges. $R$ is the number of relations in the graph and $N$ is the number of nodes in the graph. $\mathcal{N}$ is a set of nodes in graph $\mathcal{G}$. Target nodes, $\mathcal{T} \subseteq \mathcal{N}$, are nodes in the graph with class labels. $h_i^l$ is the vector representation of node $i$ at layer $l$, and $W_r$ is a relation-specific weight matrix. $k$ is the number of RGCN layers. Non-linear activation function $\sigma$ is an element-wise operation. $\mathcal{N}_r(i)$ is the collection of the incoming neighbors of node $i$ with the relation $r$.

**Theorem 2.** *For a k-layer RGCN model, the input representation $h_m^0$ of a node $m$ more than k hops away from any node in $\mathcal{T}$ is not involved in the computation of $h_i^k$ for any node $i \in \mathcal{T}$.*

*Proof.* We will prove this by induction on $k$.

**Base case (k = 1):** Let $m$ be a node which is more than 1 hop away from all nodes in $\mathcal{T}$. We need to show that the computation of $h_i^1$, $i \in \mathcal{T}$ does not involve $h_m^0$.

By definition, $h_i$ is computed as

$$\mathbf{h}_i^1 = \sigma \left[ \sum_r^R \sum_{j \in \mathcal{N}_r(i)} \frac{1}{|\mathcal{N}_r(i)|} W_r h_j^0 \right].$$

The only node representations involved are $h_j^0$ for $j \in \mathcal{N}_r(i)$, which are in the 1-hop neighborhood of $i$.

**Inductive step:** We assume that the theorem holds for $k = l$. Let node $m$ be more than $l + 1$ hops from all nodes in $\mathcal{T}$. We aim to show that for $i \in \mathcal{T}$, $h_m^0$ is not involved in the computation of $h_i$.

The output representation of $i$ is computed as

$$\mathbf{h}_i^{l+1} = \sigma \left[ \sum_r^R \sum_{j \in \mathcal{N}_r(i)} \frac{1}{|\mathcal{N}_r(i)|} W_r h_j^l \right].$$

Here $j$ is one hop from $i$ so must be more than $l$ hops from $m$ (if $j$ were $l$ hops from $m$ we could get from $i \rightarrow j \rightarrow m$ in $l + 1$ hops). From this we can apply our inductive assumption and conclude that $h_m^0$ is not involved in the computation of $h_j^l$.

We can now see that the only node representations involved in the computation of $h_i^{l+1}$ are $h_j^l$, none of which were computed using $h_m^0$.

## Notations

We use lowercase letters to denote scalars (e.g., $x$), bold lowercase letters for vectors (*e.g.*, $\mathbf{w}$), uppercase letters for matrices (e.g., $A$), and caligraphic letters for sets (*e.g.*, $\mathcal{G}$). We also use uppercase letters for referring to dimensions and lowercase letters for indexing over those dimensions (*e.g.*, $\sum_{b=1}^{B}$). The dimensions of vectors, matrices and tensors are reported in Table A1.

## General experimental setup

All experiments were performed on a single-node machine with an Intel(R) Xeon(R) Gold 5118 (2.30 GHz, 12 cores) CPU and 64 GB of RAM. GPU experiments used a Nvidia GeForce GTX 1080 Ti GPU. We used the Adam optimiser (*Kingma & Ba, 2015*) with a learning rate of 0.01. The hyperparameters for the reproduction experiements were taken from the original work by *Schlichtkrull et al. (2018)*. However, some hyperparameters were not report in the article, so we obtained the missing configurations from the original codebase and personal communication with the authors. For the new models (c-RGCN & e-RGCN), the hyperparameters were tuned using validation sets. However, we did not perform an extensive systematic hyperparameter search. For reproducibility, we provide an extension description of the hyperparameters that we have used in node classifications and link predictions in YAML files under the *configs* directory on the project GitHub page: https://github.com/thiviyanT/torch-rgcn.

We ran the original implementation of the link prediction model (https://github.com/MichSchli/RelationPrediction) on the FB15-237 dataset. The exact hyperparameters for WN18 and FB15k experiments were not available in the original codebase. We used

**Table A1 Dimensions of vectors, matrices and tensors.**

| Variables | Dimensions |
|---|---|
| $A$ | $N \times N$ |
| $X$ | $N \times F$ |
| $W$ | $N_{in} \times N_{out}$ |
| $\mathbf{h}_i$ | $1 \times N_{out}$ |
| $\mathbf{X}_i$ | $1 \times F$ or $1 \times N_{in}$ |
| $A_r$ | $R^+ \times N \times N$ |
| $W_r$ | $R^+ \times N_{in} \times N_{out}$ |
| $C$ | $R^+ \times B$ |
| $B$ | $B \times N_{in} \times N_{out}$ |
| $Q_r$ | $B \times \dfrac{N_{in}}{B} \times \dfrac{N_{out}}{B}$ |
| $A_v$ | $R^+ \times N \times N$ |
| $A_h$ | $N \times R^+N$ |
| $E$ | $N \times D$ |
| $\mathbf{W}_r$ | $1 \times N_{out}$ |
| $\mathbf{e}_s$ | $1 \times D$ |
| $\mathbf{r}$ | $1 \times D$ |
| $e_o$ | $1 \times D$ |
| $R$ | $R^+ \times D$ |
| $W_\theta$ | $C \times D$ |
| $W_\phi$ | $D \times C$ |

**Note:**
   In the order as they appear in the text.

Tensorflow 1.4, Theano 1.0.5 and CUDA 8.0.44. This replication required a Nvidia Titan RTX GPU with 24 GB of GPU memory, but model training and inference was performed on the CPU.

## Schlichtkrull initialisation

The initialization used in the link prediction models in *Schlichtkrull et al. (2018)* differs slightly from the more standard Glorot initialization (*Glorot & Bengio, 2010*).

$$std = \Phi \times \frac{3}{\sqrt{\text{fan in} + \text{fan out}}}, \qquad (17)$$

Here, gain ($\Phi$) is a constant that is used to scale the standard deviation according the applied non-linearity. *std* is used to sample random points from either a standard normal or uniform distribution. We refer to this scheme as *Schlichtkrull initialisation*. When gain is not required, $\Phi$ is set to 1.0.

## FB-Toy link prediction

We also performed link prediction experiment on the FB-Toy dataset. The mean and standard error of the link prediction results are reported in Table A2.

**Table A2  Mean and standard error of mean reciprocal rank (MRR) and Hits@k (k = 1, 3 and 10) over 10 runs for link prediction using RGCN and Torch-RGCN on the FB-Toy dataset.**

| Dataset | Model | MRR | Hits@1 | Hits@3 | Hits@10 |
|---|---|---|---|---|---|
| FB15k-Toy | TF-RGCN | 0.432 ± 0.008 | 0.293 ± 0.007 | 0.482 ± 0.011 | 0.768 ± 0.010 |
| | Torch-RGCN | 0.486 ± 0.009 | 0.352 ± 0.011 | 0.540 ± 0.009 | 0.799 ± 0.008 |

**Note:**
Triples from the truth set (train, validation and test set) have been filtered. All models that were trained on the GPU.

## ACKNOWLEDGEMENTS

We are very grateful to Michael Schlichtkrull for supporting us with the reproduction of the link prediction results. Experiments were ran on DAS-5 ASCI Supercomputer (*Bal et al., 2016*) and on the Dutch national e-infrastructure with the support of SURF Cooperative.

### Funding

This research was supported by the responsible data science track of the VSNU Digital Society program. The funders had no role in study design, data collection and analysis, decision to publish, or preparation of the manuscript.

### Grant Disclosures

The following grant information was disclosed by the authors:
VSNU Digital Society Program.

### Competing Interests

The authors declare that they have no competing interests.

### Author Contributions

- Thiviyan Thanapalasingam conceived and designed the experiments, performed the experiments, analyzed the data, performed the computation work, prepared figures and/or tables, authored or reviewed drafts of the article, and approved the final draft.
- Lucas van Berkel performed the experiments, performed the computation work, authored or reviewed drafts of the article, and approved the final draft.
- Peter Bloem conceived and designed the experiments, authored or reviewed drafts of the article, and approved the final draft.
- Paul Groth conceived and designed the experiments, authored or reviewed drafts of the article, and approved the final draft.

### Data Availability

Our RGCN implementation is available at GitHub: https://github.com/thiviyanT/torch-rgcn.

The links to third-party datasets, and their published sources, used in this work are available in the Supplemental File.

## Supplemental Information

Supplemental information for this article can be found online at http://dx.doi.org/10.7717/peerj-cs.1073#supplemental-information.

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
