# Peer review of "Relational graph convolutional networks: a closer look"

_PeerJ Computer Science, doi:10.7717/peerj-cs.1073_

## Round 0.1 · original submission · Minor Revisions

The three reviewers agree on the relevance and value of your contribution. We look forward to receiving the revised version of your work.

Reviewer 1 ·

Basic reporting

This work is well written and presented.

A small writing issue:
line 27: have become been widely adopted -> have become widely adopted

Experimental design

- "Parameter efficiency" and "Memory efficiency" are mentioned in section 3 and 4, but they are not justified with more details (quantitatively) in the experiments.
- For the task of link prediction, it would be better to have detailed results from state-of-the-art methods to quantify the gaps between RGCNs and those approaches.

Validity of the findings

The challenges of reproduction discussed in 7.2 are spot-on and convincing according this work. But arguments in 7.1 about e-RGCN and c-RGCN seems like over-claims to me. It would be better to have more quantitative results in the experimental section to support those claims.

Additional comments

Overall to me, it is a solid work which should benefit the graph learning community in general.

Reviewer 2 ·

Basic reporting

Language
=======

The manuscript is readable and well organized. However, there are several minor language issues repeated throughout:

(1) References to the literature aren't proper parenthetical expressions. This happens mostly with the main paper cited here :Schlichtkrull et al.(SKB+18). Line 79 is an example: "In (SKB+18), Schlichtkrull et al. ...". Other problematic lines are 183, 252, 275, 310, 313, 382, 402, 429, 432, 479, 497, 498, 505.

SKB+18 is cited way too many times in the paper. There are multiple paragraphs where the work is cited multiple times. It would be really nice to review the paper and carefully edit it for readability taking this into account, and minimizing the number of times that work is cited.

(2) There are many small language issues, such as faulty agreements in the text. For example, in line 61: "Relational Graph Attention Networks uses... " instead of "use". Other lines that need review include 93,
180-1, 378, 417, 419. Reference to BSCH19 appears alone between periods in line 62. On line 80, the sentence starting with "In this section..." needs work.

(3) The formalization of graphs is incorrect and inconsistent. Line 88 uses angle brackets for edges while 106 doesn't. Moreover, the stating that "<i, j> \in \cal{E} is a set of tuples" does not make sense. The customary way would be to say \cal{E} \subseteq \cal{V} \times \cal{V}. Similarly for section 3.2 which extends graphs to be edge-labelled.


Literature
=======

Section 2 serves little purpose in this paper. Starting from the premise that the paper is solely about reproducing and re-implementing previous work, the related work should be mostly about similar efforts instead of talking about tasks like node classification and link prediction.


Article Structure
============

Figures and tables are well formatted. The paper is well organized.

However, in some places there are mentions to runtime (e.g., lines 351 and 514) but the execution times for the various systems are not reported. Related to that, it is not clear that the time comparisons are fair -- there is a mention in line 492 that previous work was tested in a virtual environment. If execution time is to factored in an argument, there must be transparency with respect to how such data is collected and all such data must be disclosed and properly discussed.


Self-contained
===========

Several design decisions are made without proper justification. First and foremost, there is the "stacking trick" (Section 4.3) which is claimed to save considerable time and produce correct results. The correctness of that step should be demonstrated (proof or citation). Next, there is a mention to pruning nodes from the graph during node classification (line 304). There is a citation in that line referring to the drop in memory requirements after pruning, but the mention to no accuracy degradation is not equally addressed.

More generally, the paper provides several of the parameters and steps done in the attempt to reproduce the experiments in the SKB+18 paper but does not explain how they were tuned nor whether the same parameters were used in the original paper. For example, why are 50 epochs used in all tests except for the e-RGCN with the AM dataset (line 289). Similarly for the model setup for the link prediction task (section 6.1): are the parameters identical to the original paper? If not, why not and how are the parameters tuned?

Experimental design

Fit within scope of Journal
====================

No concerns identified.


Research questions
===============

This is an area where the paper needs work. The paper is positioned primarily as a reproduction of the original RGCN work because that code is obsolete. The paper offers another two contributions: a tutorial on the different components of RGCNs and "two new configurations of the RGCN that are more parameter efficient".

With respect to the the tutorial, claimed to provide a "friendly understanding of the different components of the RGCN", there are a few aspects that could be improved:

(1) Vector/matrix/tensor dimensions would help understanding all equations.

(2) Why are multiple convolution layers required for large graphs? (line 96) What is the relationship there? Does it depend on task?

(3) Why is single layer sufficient for a sparse graph? (line 117) What is the relationship between sparsity and layers?

(4) Why is it that having indistinguishable embeddings for different nodes undesirable? (line 119) What about clustering tasks?

(5) What are the connections between RGCNs and random walks? There are several elements in common, including the need for self-loops and trade-offs involving the length of the paths to be considered. There are several embedding methods based on random walks (e.g., the PRA work).


With respect to the paper as an exercise in reproducibility, there are issues related to why use different parameters than the original work as well as a question related to why not use other datasets that came out after the original paper. In the same way that the old code is now obsolete, so are the old datasets. More insight can be gathered by using newer and better designed datasets, and there is vast literature to that effect (e.g., Rossi et al 21 -- RBF+21).

With respect to the two new configurations, it is not clear whether or not they are a significant contribution. That could be answered by comparing them against the state-of-the-art past the original RGCN paper.


Rigor
====================

No concerns identified.


Reproducibility
===========

The paper does provide more parameters and statistics than most. However, I have not attempted to reproduce the results.

Validity of the findings

Impact and novelty
==============

This needs to broken down in parts.

As an exercise in reproducing previous work, the paper offers generally interesting insights. the main questions here concern the correctness and the utility of the new code.

For correctness, the paper mentions (line 415) experiments with a small dataset and that the results were confirmed to be correct. However, it is not clear what that really means. For example, were the results checked by hand? Were they compared to the output of the RGCN from the original paper?

The paper shows that the new code performed nearly identically to the old RGCN code for the task of node classification. On the other hand, different results were obtained for the link prediction tasks. The discrepancy is attributed to differences in tuning.

However, if the goal is to show that the new code is correct with respect to the old code, why not generate several synthetic networks with different characteristics (and sizes) and compare the new implementation against the old one under the same parameter settings?

As mentioned, the paper also offers two new configurations of RGCNs as contributions. However, the paper does not offer strong evidence that they are actual contributions to the field.


Data
====

Except for the timing data (see comments above), there are no concerns.


Conclusions
=========

Section 8 could be re-worked based on a stronger positioning of the paper. More generally, it is not crystal clear what are the contributions of the work as stated.

Additional comments

Motivation
=======

The paper should make a stronger case for why RGCNs are important and still relevant, thus justifying the work. The first two paragraphs in 7.1 do a bit of that. But it is not clear that RGCNs remain state-of-the-art in many tasks, and that should be addressed in the introduction. It would also be nice to see a discussion of the considerable follow up work following the original paper.


Training times
===========

The paper makes a suggestion that authors report training times in their papers. However, it is hard to conclude much from training/execution times unless for code that is run on the same environment. Even small changes in operating system or libraries that go unnoticed by the author can lead to drastic changes in time.

However, it is always useful to report training and execution times transparently when the authors actually run different tools, as in your case.


Reproducibility
===========

The paper makes an interesting comment on lines 528-530: "Authors should be mindful that even if legacy code remains executable for a long time, executing it efficiently on modern hardware may stop being possible much sooner."

What would be an actionable step that authors could take to address that issue?

What are you doing about your own code?

·

Basic reporting

No Comment

Experimental design

No Comment

Validity of the findings

No Comment

Additional comments

The paper was a joy to read. It clearly set out its aims and then went on to show how they were achieved. Practitioners would benefit from the re-implementation of the RGCN in this work using an updated framework. The explanation of how RGCNs work with message passing and graph convolutions also adds great pedagogical value to those interested in the field. Bonus points for the clear and colourful illustrations and accessible language.
Section 7.2 was insightful and the practical action of providing results on small graph datasets which will run quickly has potential to really make the lives of reproducers easier.
The work on c-RGCN does indeed seem promising and hopefully can spark more research along those lines to solve the problem of efficiency of complex approaches to link prediction.

I found only minor comments to make. They are below.

Questions

1. In Table 2, the e-RGCN seems to perform much worse than the others in the AIFB dataset. Lines 274-276 gave the impression that the original re-implementation (Torch-RGCN as I understand it) performed worse than e-RGCN in all aspects, can you clarify?

2. The first time c-RGCN is mentioned seems to be line 381 (Page 13), but it is not introduced until 431 (page 14).

3. It could be useful to mention that the datasets mentioned in lines 288-294 will be introduced a bit later on.

4. Are there any intuitions as to why the RGCN is poor at link prediction?

Grammatical
1. Page 2, line 52 "... used learned to entities ...".
2. Page 13, lines 373-374: "... which is not be applicable to Knowledge Graphs.".
3. Page 15, lines 425-427, the sentence seems incomplete: "We consider this ... state-of-the-art KGE models.".
4. Page 16, line 462: "... models and is good starting place...".

---

## Round 0.2 · Minor Revisions

Thank you for your contribution to PeerJ Computer Science and for systematically addressing all the reviewers' suggestions. The reviewers are satisfied with the revised version. However, there is a minor request from reviewer 2 regarding citation format. This should be easy to fix. Once this is addressed the manuscript will receive the final "accept" decision. Congratulations in advance! We look forward to receiving the revised version of your work.

Reviewer 1 ·

Basic reporting

This work is well written and presented.

Experimental design

My previous comments: 1) "Parameter efficiency" and "Memory efficiency" are mentioned in section 3 and 4, but they are not justified with more details (quantitatively) in the experiments. 2) For the task of link prediction, it would be better to have detailed results from state-of-the-art methods to quantify the gaps between RGCNs and those approaches.

Generally, the authors soften their claims which make this work more scientifically rigorous. However, it also somewhat weaken the contributions. Instead, I'd love to see some more experimental results to support the original claims.

Validity of the findings

My previous comments: The challenges of reproduction discussed in 7.2 are spot-on and convincing according to this work. But arguments in 7.1 about e-RGCN and c-RGCN seems like over-claims to me. It would be better to have more quantitative results in the experimental section to support those claims.

Likewise, the authors soften their claims which is good to see. But more experiments would be even better.

Additional comments

All my previous comments are addressed by softening the original claims, though no additional experiment is added. As a result, the results appear to be less impressive. Still a solid work and more rigorous which should benefit the graph learning community in general. Hence an accept.

Reviewer 2 ·

Basic reporting

Overall the authors did a good job with the revisions. The article is clear and understandable without much effort and the discussion of related work is now much clearer and appropriate.

There is still no consistency with the citation to the literature, which goes strongly against the notion that the the article uses professional English as stipulated by the guidelines. Here are just some examples, in the interest of time:

line 57: (Tatman et al. 2018) propose a simple... --> sentence started with parenthetical expression
line 60: Similarly, (Heil et al. 2021) propose standards... -> clause started with parenthetical expression
line 83: Daza and Cochez (Daza & Cochez 2020) have explored... -> bad style
line 92: We refer the reader to (Ruffinelli et al. 2019) and ...

There are many citations that are correct in the paper. The authors seem to struggle only when wanting to refer to the authors of the cited work, in which case only the year of the publication goes inside the parenthesis.

Experimental design

The article describes its methods and evaluation criteria clearly. It also explains the goals of the experiments and interprets the results appropriately.

Validity of the findings

One concern raised in the first submission was that the authors used earlier benchmark datasets for some tasks even though they were known to be flawed. The revised article does not address the concern, but since the article is about reproducing an existing method, this is less of a concern.

Additional comments

Overall this is a much improved manuscript. The work is interesting on its own. Hopefully the writing can be improved to make it for a more readable artile.

·

Basic reporting

No comments.

Experimental design

No comments.

Validity of the findings

No comments.

Additional comments

I am happy with the changes made and the authors' responses to the issues raised by the reviewers.

---

## Round 0.3 · accepted · Accept

Thank you for promptly taking care of the requested changes by correcting the citations. Thank you also for carefully revising your manuscript and fixing some minor typos.